# Sex differences in long-term survival after intensive care unit treatment for sepsis: A cohort study

Kelly Thompson[1,2]*, Naomi Hammond[1], Michael Bailey[3], Jai Darvall[4,5], Gary Low[1,2,6], Steven McGloughlin[3,7], Lucy Modra[4,8], David Pilcher[3,6,9]

**1** The George Institute for Global Health, UNSW Sydney, Newtown, NSW, Australia, **2** Nepean and Blue Mountains Local Health District, Kingswood, NSW, Australia, **3** Australian and New Zealand Intensive Care Research Centre, Monash University, Melbourne, Australia, **4** Department of Critical Care, University of Melbourne, Melbourne, Australia, **5** Department of Intensive Care Medicine, Royal Melbourne Hospital, Melbourne, Australia, **6** Sydney Medical School, Faculty of Medicine and Health, University of Sydney, Sydney, Australia, **7** Department of Intensive Care, Alfred Health, Prahran, Melbourne, Australia, **8** Intensive Care Unit, Austin Health, Melbourne, Australia, **9** The Australian and New Zealand Intensive Care Society (ANZICS) Centre for Outcome and Resource Evaluation, Camberwell Road, Camberwell, Melbourne, Australia

* kthompson@georgeinstitute.org

**Data Availability Statement:** The data underlying the results presented in the study are available from ANZICS CORE upon request https://www.anzics.com.au/core-portal/.

## Abstract

### Objective

To determine the effect of sex on sepsis-related ICU admission and survival for up to 3-years.

### Methods

Retrospective cohort study of adults admitted to Australian ICUs between 2018 and 2020. Men and women with a primary diagnosis of sepsis were included. The primary outcome of time to death for up to 3-years was examined using Kaplan Meier plots. Secondary outcomes included the duration of ICU and hospital stay.

### Results

Of 523,576 admissions, there were 63,039 (12·0%) sepsis-related ICU admissions. Of these, there were 50,956 patients (43·4% women) with 3-year survival data. Men were older (mean age 66·5 vs 63·6 years), more commonly received mechanical ventilation (27·4% vs 24·7%) and renal replacement therapy (8·2% vs 6·8%) and had worse survival (Hazard Ratio [HR] 1·11; 95% Confidence Interval [CI] 1·07 to 1·14, P<0·001) compared to women. The duration of hospital and ICU stay was longer for men, compared to women (median hospital stay, 9.8 vs 9.4 days; p<0.001 and ICU stay, 2.7 vs 2.6 days; p<0.001).

### Conclusion

Men are more likely to be admitted to ICU with sepsis and have worse survival for up to 3-years. Understanding causal mechanisms of sex differences may facilitate the development of targeted sepsis strategies.

**Funding:** KT received funding for the work from the National Health and Medical Research Council of Australia. The funders had no role in study design, data collection and analysis, decision to publish, or preparation of the manuscript.

**Competing interests:** The authors have declared that no competing interests exist.

## Introduction

Sepsis is life-threatening organ dysfunction that occurs due to the hosts dysregulated response to infection [1]. It is the primary cause of death from infection, especially if not recognised and treated promptly [1]. In 2017, there was an estimated 49 million sepsis cases and 11 million sepsis-related deaths, accounting for approximately one-fifth of all deaths globally [2, 3].

Three systematic reviews of sex differences in mortality of sepsis patients treated in the ICU have reported an absence of well-designed studies and significant heterogeneity [4–6]. In studies where sex differences in sepsis mortality are observed, differences between men and women's innate and adaptive immune response are cited [7], with greater protection in women attributed to the immune enhancing effects of estrogen, and male disadvantage related to the immunosuppressive properties of male androgens, particularly testosterone. Other possible causes include gender differences in health seeking behaviours [8] and gender bias associated with the delivery of healthcare [9].

The objective of this study was to determine the effect of sex on sepsis-related ICU admission and long-term survival for up to 3-years in Australian ICUs.

## Materials and methods

### Design, setting and participants

We conducted a retrospective cohort study of Australian ICU admissions between January 1, 2018 and December 31, 2020 using data from the Australia and New Zealand Intensive Care Society (ANZICS) Adult Patient Database (APD) [10]. The APD is a clinical quality registry that includes detailed information on more than 90% of all ICU admissions in Australia and New Zealand, managed by the ANZICS Centre for Outcome and Resource Evaluation (CORE). The APD is used for routine quality-assurance benchmarking processes with data collected by trained data collectors in participating ICUs. For this study, data from the APD were linked to the National Death Index by the data linkage unit of The Australian Institute for Health and Welfare using a statistical linkage key (SLK-581) which was introduced into routine ANZICS data collection in 2017 [11]. De-identified linked data was then provided back to the researchers. The study (Sex differences in sepsis prevalence and outcomes from the Australian and New Zealand Intensive Care Society Admitted Patient Database) was approved on the 8th of April 2021 by the Alfred Hospital Human Research Ethics Committee (HREC 270–21) with a waiver of consent. All procedures were followed in accordance with the ethical standards of the responsible committee on human experimentation and with the Helsinki Declaration of 1975.

Sepsis was identified using a modified version of the 2016 consensus definition of sepsis [1]. All patients admitted to the ICU with a primary diagnosis of suspected or confirmed infection who also had a Sequential Organ Failure Assessment score of 2 or more within 24 hours of admission to ICU were included [12]. Infection-related diagnoses at the time of admission were classified according to the ANZICS modification of the Acute Physiology and Chronic Health Evaluation (APACHE) [13] III scoring system and used to infer the presence of suspected or confirmed infection (S1 Table in S1 File). Criteria for sepsis were assessed within the first 24 hours of ICU admission.

Participant sex was defined based on the sex recorded in the APD [14]. In the APD sex is transcribed from the clinical record with biological sex (male/female/intersex or indeterminate sex/unknown) recorded. It has been previously noted that the terms sex and gender are used interchangeably in the critical care literature [15]. For this study we refer to males as men and females as women throughout the manuscript.

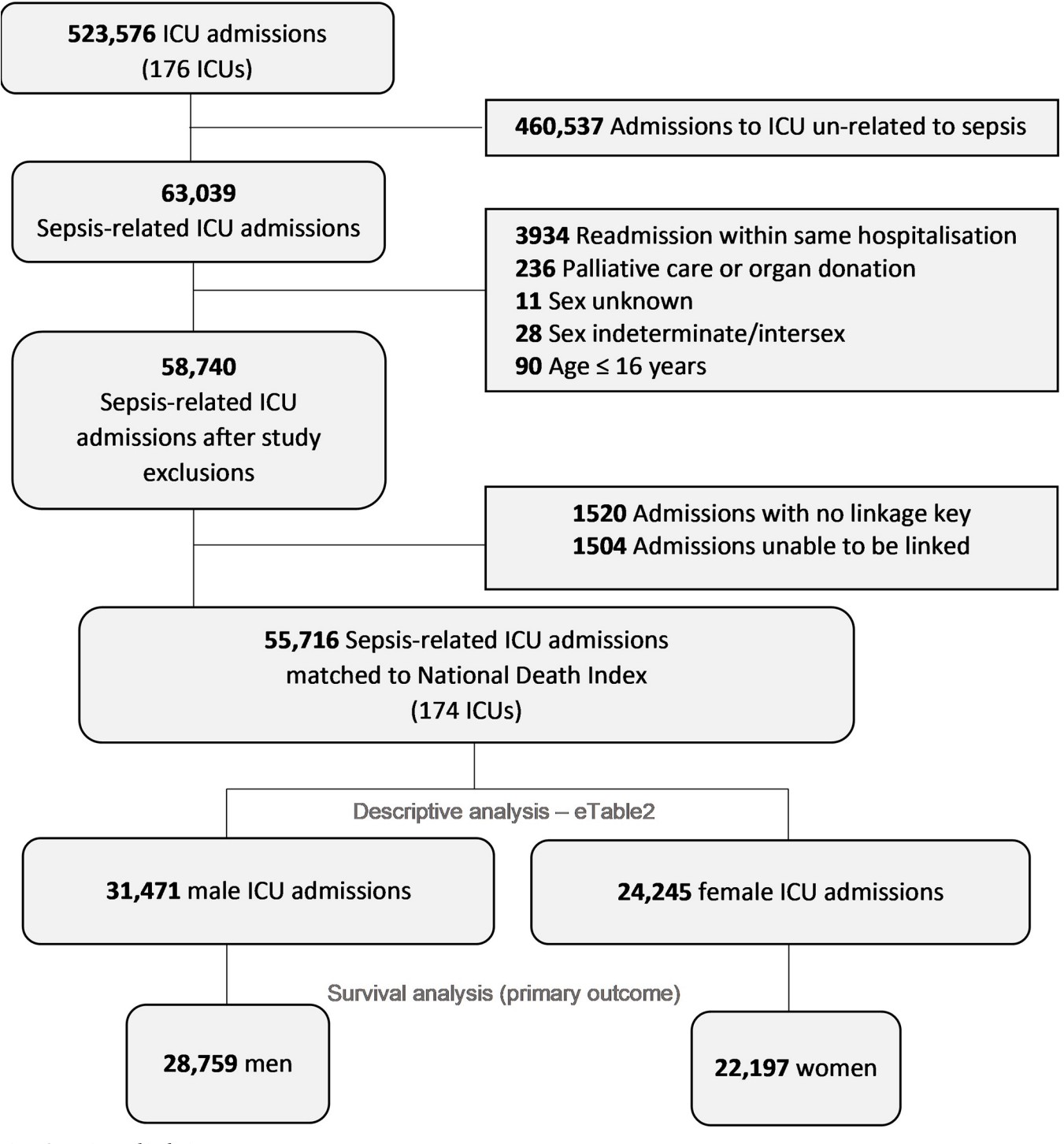

**Fig 1. Screening and inclusion.**

Patients aged younger than 16 years, those in whom there was no sex variable recorded, those in whom there were no outcome data available and subsequent readmissions to ICU within the same hospital admission were excluded. Due to small numbers (<0·1%), those listed as intersex/indeterminate sex were also excluded (Fig 1).

Acute illness severity was assessed using the Acute Physiology Score of the APACHE III/IV scoring system and individual lactate levels at presentation to ICU. Comorbidities were assessed using the chronic health variables of the APACHE III/IV scoring system [13, 16]. Frailty was assessed using a modification to the nine-point Canadian Study of Health and Aging Clinical Frailty Scale (CFS), an assessment tool used to quantify frailty based on the deficit accumulation approach [17]. In the APD, the CFS is modified to eight-categories without a CFS of 9 (terminally ill) [18]. In this study we classified frailty as follows; no frailty (<5), mild to moderate frailty (5 to 6) and severe to very severe frailty (7 to 8).

Patients were grouped into the following diagnostic categories 1. sepsis source unknown or not specified, 2. gastro-intestinal, 3. neurological, 4. renal/urinary/gynaecological, 5. respiratory and 6. skin and soft-tissue. Socioeconomic status was assessed by linking the index of social advantage and disadvantage [19] to the patient's postcode and reporting data by quintiles. States and territories of Australia were collapsed into four regions (1. New South Wales / Australian Capital Territory, 2. Queensland, 3. South Australia / Western Australia / Northern Territory, 4. Victoria / Tasmania).

### Outcomes

The primary outcome was time to death from ICU admission for up to three-years. This endpoint was assessed by sex, overall, and in subsequent sensitivity analysis by excluding those who died in hospital. The following age groups were also examined: <55 years, 55 to 74 years, ≥ 75 years. We also included patients with probable or confirmed Severe Acute Respiratory Syndrome Coronavirus 2 (SARS-CoV-2) as a subgroup. Sex-specific risk factors for death following a sepsis ICU admission were also assessed. Secondary outcomes included in-ICU mortality, in-hospital mortality, rate of readmission to ICU within the same hospital stay and length of stay in ICU and in hospital.

### Statistical analysis

Data are presented as percentages and numbers, means with standards deviations (SD), medians and interquartile ranges (IQR), or proportions and 95% confidence intervals (CI). To test differences, we used $\chi^2$ tests for equal proportion, *t* tests, or Wilcoxon rank sum tests accordingly. Probability of survival up to three-years after ICU admission was examined using Kaplan Meier plots for the overall study cohort, in sex-specific age groups and in those with probably or confirmed SARS-CoV-2. Cox proportional hazard models were used to estimate the effect of sex on time to death following a sepsis ICU admission adjusting for age, diagnosis, acute illness severity, body mass index, chronic comorbidities, frailty, source of admission to ICU and to hospital, treatment limitation at ICU admission, socio-economic status, region of Australia, type of hospital (regional/rural, metropolitan, tertiary or private) and year of admission to ICU. To account for sex-specific differences in risk factors, we created individual Cox regression models for men and women to identify the independent impact of confounding variables on survival, and compare relative hazard ratios in men to women, where a ratio greater than one indicated that the variable in question was associated with an increased risk of death for men compared to women with the results reported in a forest plot with 95% confidence intervals (CI) [20]. P-values of <0.05 and 95% CI that did not cross the value of 1 were considered statistically significant. All analyses were conducted using Stata version 16·1 (Texas, USA).

### Results

Of 523,576 admissions from 170 ICUs, there were 63,039 (12·0%) sepsis-related ICU admissions. Of these, 50,956 patients, 28,759 (56·4%) men and 22,197 (43·4%) women with available

long-term survival data were identified in administrative datasets held by Australian Institute of Health and Welfare (Fig 1).

The baseline characteristics and outcomes of patients with sepsis are presented by sex in Table 1. The baseline characteristics of all sepsis-related ICU admissions overall and by sex are reported in S2 Table in S1 File with characteristics of those who could not be matched (n = 3024) reported in S3 Table in S1 File. At admission to ICU, men were older (mean age 66.5 vs 63.6 years) and had higher APACHE II (19·2 vs 18·8), APACHE III (65·4 vs 62·8), SOFA (5·6 vs. 5·2) and ANZROD (14·9 vs 13·6) scores compared to women. Men more commonly received mechanical ventilation (27·4% vs 24·7%) and renal replacement therapy (8·2% vs 6·8%) compared to women. Women had a higher incidence of baseline frailty at ICU admission (CFS 5–8; 23·9% vs. 21·2%). There were similar proportions of men and women with a diagnosis of septic shock (51·2% in men vs 50·4% in women).

## Length of stay and survival

The duration of hospital and ICU stay was longer for men, compared to women (median hospital stay, 9.8 days vs 9.4 days; p<0.001 and ICU stay, 2.7 days vs 2.6 days; p<0.001). A higher proportion of men died in hospital during the sepsis admission, compared to women (4277 [14.8%] vs 2896 [13.0%]; p<0.001) (Table 1).

Overall women had better survival for up to three-years after ICU admission (P<0·001) compared with men (Fig 2). In all three age groups (<55 years, 55 to 74 years, ≥ 75 years) women had better survival than men (S1 Fig, Panel B, C, D in S1 File). We saw similar survival trends in the subgroup of those with probable or confirmed SARS-CoV-2 (S3 and S4 Figs in S1 File). After adjusting for confounding factors, men had a significantly higher risk of death, compared to women for up to 3-years (HR 1·11; 95% CI 1·07 to 1·14, P<0.001). Similar trends were observed when looking at the sensitivity analyses which included those who survived the hospital admission overall and by prespecified age groups (S2 Fig in S1 File).

## Sex-specific effects of predictor variables on survival time

When examining the individual sex-specific effects of predictor variables on survival time, there were no significant differences between men and women with a gastrointestinal source of sepsis. Neurological (HR 1·17; 95% CI 1·14 to 1·21), renal/urinary/gynaecological (HR 1·04; 95% CI 1·03 to 1·05), respiratory (HR 1·16; 95% CI 1·14 to 1·17) and soft-tissue (HR 1·21; 95% CI 1·17 to 1·24) sources of sepsis were associated with a higher risk of death in men compared to women (Fig 3). Body Mass Index of 30–34·9 was associated with a higher risk of death in women compared to men (HR 0·82; 95% CI 0·8 to 0·85). The presence of metastatic cancer was associated with a significantly shorter time to death in women compared to men (HR 0·80; 95% CI 0·79–0·82). The effect of predictor variables on survival time by sex and comparing men to women are reported in Fig 4.

There were no sex differences in survival when assessing the relative impacts of the source of admission to hospital and ICU, the presence of a treatment limitation, hospital type, region and socio-economic status. Results of the full cox proportional hazard regression models overall, in men and in women are reported in S4 Table in S1 File.

## Discussion

Among more than 50,000 sepsis ICU admissions in Australia, male sex was associated with an increased risk of ICU admission. Men had worse survival rates for up to 3-years, compared to women. Neurological, renal/urinary/gynaecological, respiratory and skin or soft-tissue related

**Table 1. Characteristics of men and women admitted with sepsis.**

| Characteristics | Men | | Women | |
|---|---|---|---|---|
| | **28,759** | | **22,197** | |
| Age, mean (SD) | 66·5 | 15·8 | 63·6 | 17·3 |
| APACHE II score, mean (SD) | 19·2 | 7·6 | 18·8 | 7·5 |
| APACHE III score, mean (SD) | 65·4 | 24·6 | 62·8 | 24·9 |
| SOFA score day 1, mean (SD) | 5·6 | 2·9 | 5·2 | 2·7 |
| ANZROD, mean (SD) | 14·9 | 19·43 | 13·6 | 18·8 |
| **Source of sepsis,** No. (%) | | | | |
| Gastro-intestinal | 3,662 | 12·7 | 2,754 | 12·4 |
| Neurological | 652 | 2·3 | 549 | 2·5 |
| Other/unknown | 10,984 | 38·2 | 7,827 | 35·2 |
| Renal/urinary/gynae | 3,573 | 12·4 | 3,851 | 17·3 |
| Respiratory | 7,852 | 27·3 | 5,783 | 26·0 |
| Skin & soft-tissue | 2,030 | 7.0 | 1,439 | 6·5 |
| **Septic shock,** No. (%) | 14501 | 50·4 | 11383 | 51·0 |
| Medical admission, No. (%) | 24972 | 86·8 | 19460 | 87·6 |
| **Body mass index (BMI)** | | | | |
| BMI <18.5, No. (%) | 433 | 1·5 | 535 | 2·4 |
| BMI 18.5 to 24.9, No. (%) | 3,954 | 13·7 | 3,077 | 13·9 |
| BMI 25 to 29.9, No. (%) | 4,672 | 16·2 | 2,756 | 12·4 |
| BMI 30 to 34.9, No. (%) | 2,729 | 9·5 | 1,846 | 8·3 |
| BMI 35+, No. (%) | 2,516 | 8·7 | 2,621 | 11·8 |
| BMI missing, No. (%) | 14,455 | 50·3 | 11,362 | 51·2 |
| **Chronic comorbidity** No. (%) | | | | |
| Respiratory disease | 2857 | 9·9 | 2307 | 10·4 |
| Cardiac disease | 3166 | 11·0 | 1953 | 8·8 |
| Liver disease (cirrhosis) | 743 | 2·6 | 471 | 2·1 |
| Kidney disease (dialysis dependent) | 1870 | 6·5 | 1191 | 5·4 |
| Immunosuppressed (therapy) | 1812 | 6·3 | 1327 | 6·0 |
| Immuno-suppressed (disease) | 3519 | 12·2 | 2799 | 12·6 |
| History of lymphoma | 759 | 2·6 | 442 | 2·0 |
| History of metastases | 1721 | 5·9 | 1115 | 5·0 |
| History of leukaemia | 1187 | 4·1 | 671 | 3·0 |
| **Clinical Frailty Score** | | | | |
| Not frail (CFS 1–4), No. (%) | 13,193 | 45·9 | 9,883 | 44·5 |
| CFS (5–6), No. (%) | 4,848 | 16·9 | 4,166 | 18·8 |
| CFS (7–8), No. (%) | 1,274 | 4·4 | 1,149 | 5·2 |
| CFS unknown, No. (%) | 9,444 | 32·8 | 6,999 | 31·5 |
| **Clinical characteristics within 24 hours of admission (SD)** | | | | |
| Highest temperature (˚C), mean (SD) | 37·56 | 1·0 | 37·5 | 1.0 |
| Highest heart rate (bpm), mean (SD) | 105·91 | 23·8 | 106 | 23.0 |
| Lowest mean arterial pressure (mmHg), mean (SD) | 63·89 | 11·10 | 62 | 11.0 |
| **Laboratory characteristics within 24 hours of admission** | | | | |
| Highest white cell count (x$10^9$/L), mean (SD) | 15·24 | 12·8 | 15·3 | 11·6 |
| **Lactate (mmol/L)** | | | | |
| Lactate <2 (%), No. (%) | 12,249 | 42·6 | 10,122 | 45·6 |
| Lactate 2 to 3.9, No. (%) | 7,413 | 25·8 | 5,141 | 23·2 |
| Lactate 4 to 5.9, No. (%) | 1,798 | 6·3 | 1,245 | 5·6 |

*(Continued)*

**Table 1.** (Continued)

| Characteristics | Men | | Women | |
|---|---|---|---|---|
| | **28,759** | | **22,197** | |
| Lactate 6 to 7.9, No. (%) | 687 | 2·4 | 534 | 2·4 |
| Lactate 8 to 9.9, No. (%) | 376 | 1·3 | 260 | 1·2 |
| Lactate 10 to 11.9, No. (%) | 210 | 0·7 | 164 | 0·7 |
| Lactate 12 to 13.9, No. (%) | 137 | 0·5 | 114 | 0·5 |
| Lactate 14+, No. (%) | 235 | 0·8 | 210 | 0·9 |
| Lactate missing, No. (%) | 5,654 | 19·7 | 4,407 | 19·9 |
| Creatinine (μmol/L), mean (SD) | 172·07 | 161·1 | 138·23 | 141·3 |
| Bilirubin (μmol/L), mean (SD) | 26·32 | 50·8 | 21·40 | 45·0 |
| **Organ supportive therapies No. (%)** | | | | |
| Inotropes/vasopressor use | 14746 | 57·7 | 11,121 | 56·6 |
| Invasive mechanical ventilation | 7020 | 27·4 | 4871 | 24·7 |
| Renal replacement therapy | 2239 | 9·2 | 1446 | 7·7 |
| **Source of Hospital admission No. %** | | | | |
| Home | 21,758 | 75·7 | 16,765 | 75·5 |
| Other acute hospital (not ICU) | 5,595 | 19·5 | 4,334 | 19·5 |
| Nursing home / chronic care / palliative care | 486 | 1·7 | 450 | 2·0 |
| Other hospital ICU | 496 | 1·7 | 331 | 1·5 |
| Rehabilitation facility | 179 | 0·6 | 142 | 0·6 |
| Other (incl. mental health, inborn & unknown) | 245 | 0·9 | 175 | 0·8 |
| **Source of ICU admission No. %** | | | | |
| Operating theatre/recovery | 3,822 | 13·3 | 2,749 | 12·4 |
| Emergency department | 14,043 | 48·8 | 10,921 | 49·2 |
| Ward | 8,150 | 28·3 | 6,279 | 28·3 |
| ICU same hospital | 21 | 0·1 | 18 | 0·1 |
| Other hospital (incl. ICU) | 2,676 | 9·3 | 2,198 | 9·9 |
| Other / unknown | 47 | 0·2 | 32 | 0·1 |
| **Hospital outcomes** | | | | |
| Died in hospital No. % | 4277 | 14·8 | 2896 | 13·0 |
| Died in ICU No. % | 2602 | 9·0 | 1791 | 8·0 |
| ICU length of stay in days | 2·7 | 1·4–4·9 | 2.6 | 1·4–4.7 |
| Median (IQR) | | | | |
| Hospital length of stay in days | 9·8 | 5·4–18·8 | 9.4 | 5.2–17.9 |
| Median (IQR) | | | | |
| **Index of Relative Advantage and Disadvantage**[**] | | | | |
| Lowest quintile | 5,910 | 20·6 | 4,529 | 20·4 |
| Second lowest quintile | 5,941 | 20·7 | 4,595 | 20·7 |
| Middle quintile | 5,844 | 20·3 | 4,471 | 20·1 |
| Second highest quintile | 5,067 | 17·6 | 3,973 | 17·9 |
| Highest quintile | 5,752 | 20·0 | 4,473 | 20·1 |
| Unknown (no postcode) | 245 | 0·9 | 156 | 0·7 |

Abbreviations: ICU = intensive care unit, SD = standard deviation, BMI = body mass index, APACHE = Acute Physiology and Chronic Health Evaluation, SOFA = Sequential Organ Failure Assessment, ANZROD = Risk of Death, CFS = Clinical Frailty Score, μmol/L = micromoles per litre, mmol/L = millimoles per litre

[**]The index of disadvantage used was the Socio-Economic Indexes for Areas (SEIFA). SEIFA consists of four indexes. For this study we used the Index of Relative Socio-Economic Disadvantage (IRSD). The IRSD scores each area by summarising attributes of the population, such as low income, low educational attainment, high unemployment and jobs in relatively unskilled occupations. It reflects the overall or average level of disadvantage of the population of an area and is a general socio-economic index that summarises a range of information about the economic and social conditions of people and households within an area.

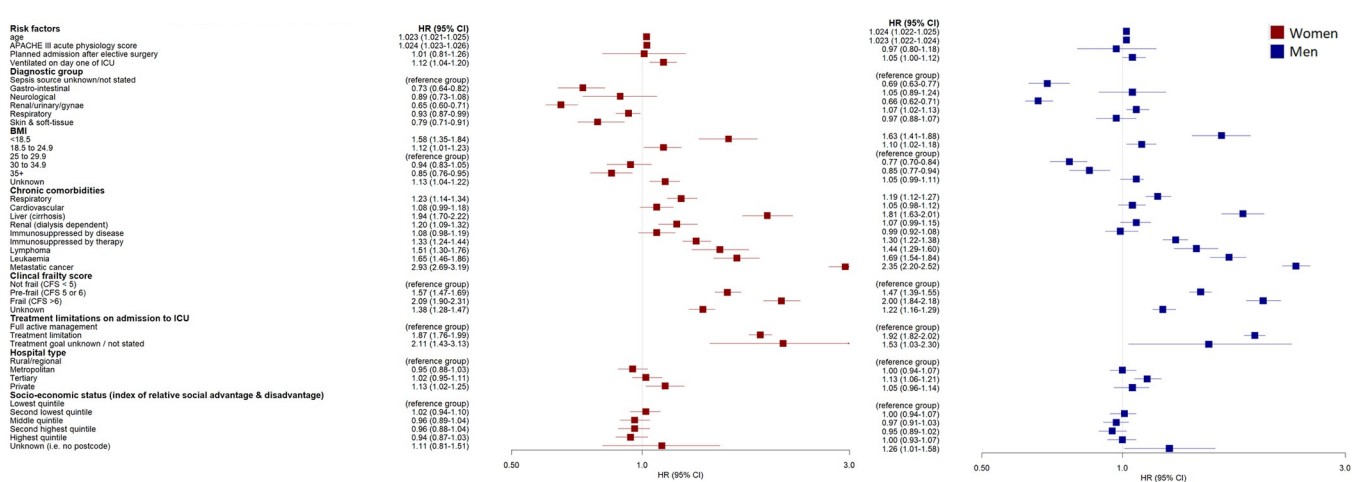

**Fig 2. Sepsis survival at 3 years in women and men overall.**

sepsis diagnoses were associated with significantly shorter survival times in men, compared to women. Women with metastatic cancer had significantly shorter survival times compared to men with metastatic cancer.

**Fig 3. Sex specific effects of predictor variables on survival in women and men.**

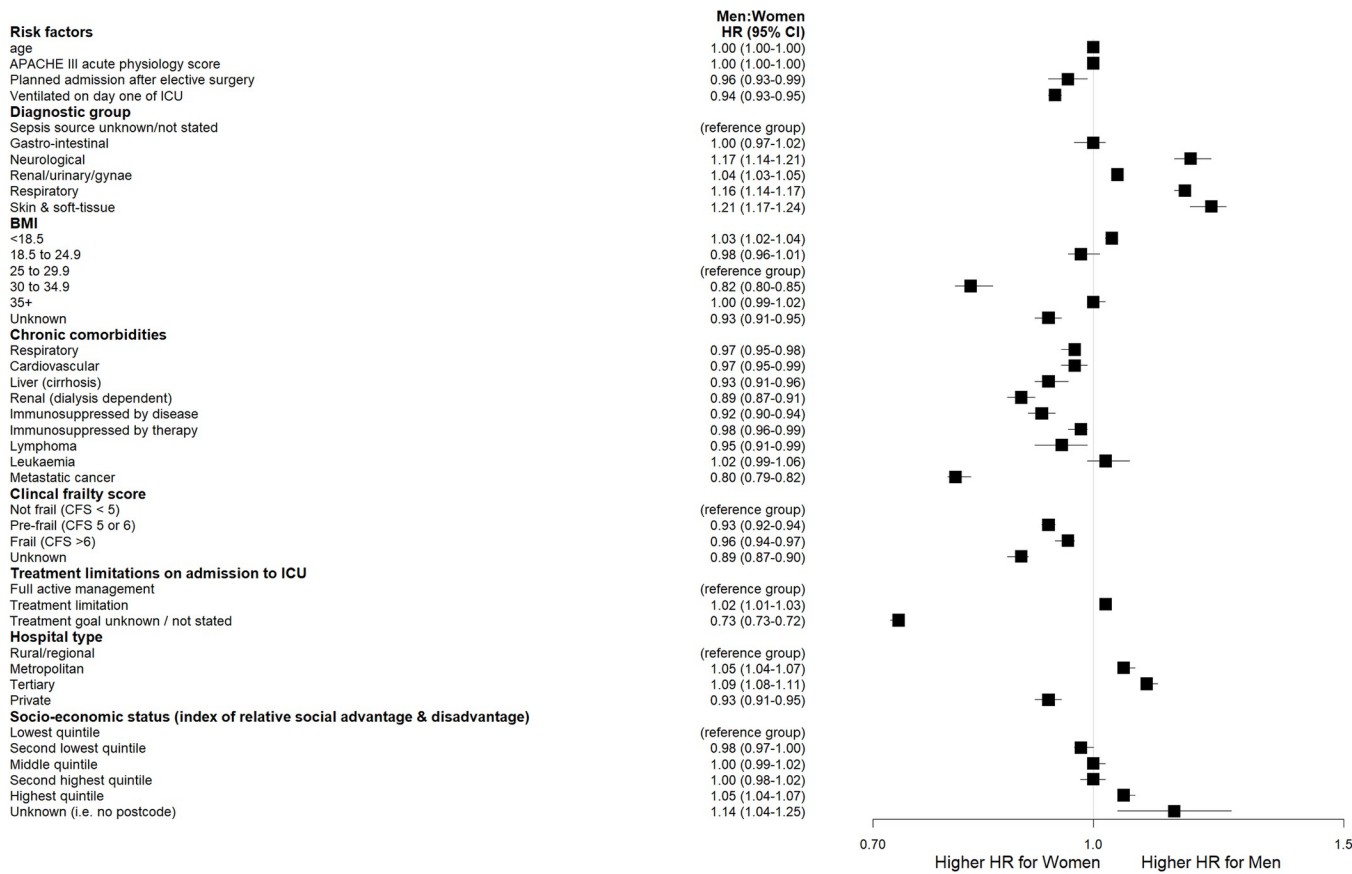

**Fig 4. Sex specific effects of predictor variables on survival comparing men wo women.**

These findings are similar to those from a recent systematic review including 71,850 patients from 12 studies that reported a survival advantage for women at one year [21]. A 2021 retrospective review of 12,321 ICU records in Boston reported similar overall findings of higher mortality due to sepsis in men compared to women overall at one year, but no difference in short and longer-term survival for men and women aged over 50 years [22]. The differences in mortality observed between our study and the Boston study may be due to our longer study follow-up time of up to 3-years. Our results further emphasise the need for targetted research to understand the causal mechanisms of differences between men and women in the severity of sepsis at ICU admission and longer term survival, and the need for awareness and action to address this disparity.

Possible explanations for the sex differences in sepsis related ICU admissions and outcomes observed in this study include biological causes [23]. Greater protection in women may be attributed to the immune enhancing effects of estrogen, while male disadvantage may be attributed to the immunosuppressive properties of male androgens, most notably testosterone [24] and 5α-dihydrotestosterone (DHT). These findings are parallel to evidence related to illness severity and outcomes from the coronavirus pandemic, where men are more likely to be hospitalised, admitted to the ICU and have higher mortality, compared to women [25]. Together these findings highlight the importance of routinely considering sex in study design and reporting of results when assessing the relative safety and effectiveness of proposed sepsis treatments, including ensuring equal numbers of men and women are recruited to trials [25–27].

The observed sex difference in survival may also reflect inequity of access to the ICU between men and women. For example, clinicians may apply a higher ICU admission threshold for men compared to women, reflected in the higher illness severity scores in men at baseline. This paper has not examined the denominator of all hospital patients with sepsis, though a recent longitudinal cohort study of older adults living in New South Wales, Australia reported similar findings, with men more likely to have a sepsis-related hospitalisation and a higher risk of death at one-year, compared to women, suggesting gender bias in clinician's selection for ICU admission is unlikely [28]. Similarly, higher numbers of men admitted to ICU with sepsis may indicate that men are more likely to delay seeking treatment for infection. In a sample of the same study population, higher illness severity scores on presentation were associated with reduced long-term survival for out to five years [29]. Whether men and women exhibit different sepsis symptoms is unknown and is an important area of future research that may lead to better targeted health promotion strategies.

Research into sex differences in health and disease has grown exponentially in the past two decades leading to greater awareness of the underrepresentation of women in clinical research along with other underrepresented racial and ethnic groups [30]. Despite an increasing focus on the need for greater diversity in research, there remains a large evidence gap in how to translate evidence of difference into a nuanced understanding of causal mechanisms of difference and subsequent change in clinical practice. The development and embedment of principles of inclusive clinical research into local, national and global research frameworks, such as the Surviving Sepsis Campaign Guidelines, would help to ensure the goals of inclusivity are prioritised and support improved patient outcomes [30].

This study has several limitations. First, the recent introduction of intersex/non-binary classifications into the APD meant that this group of patients accounted for <0·1% of ICU admissions. We therefore had to limit the classification of sex in this study to a binary definition (men/women). Second, ethnicity is not collected in the APD and therefore we were unable to take an intersectional approach to analysing data and interpreting the results. Third, this study only included patients with sepsis requiring ICU treatment, and therefore missed capturing sepsis occurring outside of the ICU or acquired during ICU stay. Fourth, the 3204 ICU patients who were unable to be matched to the national death index were younger and had lower illness severity at admission, meaning the true survival for all sepsis patients treated in the ICU may be better than reported. However, there is no evidence to indicate that this would impact the findings of relative effects in men and women. Finally, detailed information on therapies provided while in the ICU was lacking and as with any observational study, we were unable to account for unknown or other relevant confounding factors.

## Conclusion

Compared to women, men have an increased risk of sepsis related ICU admission and worse survival rates for up to 3-years. Future research to determine the underlying causes of these differences may improve awareness of sex-specific risk factors for sepsis, leading to more personalised recognition, treatment and management strategies.

## Supporting information

**S1 File.**
(DOCX)

## Acknowledgments

The authors and the ANZICS CORE management committee would like to thank clinicians, data collectors and researchers at the following contributing sites:

Royal Darwin Hospital ICU, Bathurst Base Hospital ICU, Alice Springs Hospital ICU, Canberra Hospital ICU, Albury Wodonga Health ICU, Ashford Community Hospital ICU, The Queen Elizabeth (Adelaide) ICU, Royal North Shore Hospital ICU, Warringal Private Hospital ICU, Royal Perth Hospital ICU, Royal Prince Alfred Hospital ICU, Box Hill Hospital ICU, St George Hospital (Sydney) ICU, Austin Hospital ICU, Fairfield Hospital ICU, Royal Brisbane and Women's Hospital ICU, Brisbane Private Hospital ICU, Dandenong Hospital ICU, Greenslopes Private Hospital ICU, Cairns Hospital ICU, Knox Private Hospital ICU, Ballarat Health Services ICU, John Hunter Hospital ICU, Calvary Wakefield Hospital (Adelaide) ICU, Gosford Hospital ICU, Epworth Hospital (Richmond) ICU, Bendigo Health Care Group ICU, The Valley Private Hospital ICU, Mater Adults Hospital (Brisbane) ICU, Redcliffe Hospital ICU, Sutherland Hospital & Community Health Services ICU, Mater Private Hospital (Brisbane) ICU, Flinders Medical Centre ICU, Liverpool Hospital ICU, Coffs Harbour Health Campus ICU, The Prince Charles Hospital ICU, Lingard Private Hospital ICU, Ryde Hospital, Prince of Wales Hospital (Sydney) ICU, Concord Hospital (Sydney) ICU, Goulburn Valley Health ICU, Orange Base Hospital ICU, Rockhampton Hospital ICU, Northeast Health Wangaratta ICU, Sunnybank Hospital ICU, Calvary Hospital (Lenah Valley) ICU, St John Of God Hospital (Ballarat) ICU, Royal Adelaide Hospital ICU, Lismore Base Hospital ICU, Alfred Hospital ICU, Blacktown Hospital ICU, Nepean Hospital ICU, Launceston General Hospital ICU, Sydney Adventist Hospital ICU, Calvary Mater Newcastle ICU, Lyell McEwin Hospital ICU, Toowoomba Hospital ICU, Tweed Heads District Hospital ICU, Port Macquarie Base Hospital ICU, Hornsby Ku-ring-gai Hospital ICU, St Andrew's Hospital (Adelaide) ICU, Logan Hospital ICU, Mackay Base Hospital ICU, Bankstown-Lidcombe Hospital ICU, St John Of God Hospital (Murdoch) ICU, Wollongong Hospital ICU, St George Private Hospital (Sydney) ICU, Cabrini Hospital ICU, Monash Medical Centre-Clayton Campus ICU, Prince of Wales Private Hospital (Sydney) ICU, Royal Hobart Hospital ICU, North Shore Private Hospital ICU, Westmead Hospital ICU, The Northern Hospital ICU, University Hospital Geelong ICU, Mater Private Hospital (Sydney) ICU, Wagga Wagga Base Hospital & District Health ICU, North West Regional Hospital (Burnie) ICU, Gold Coast University Hospital ICU, St Vincent's Hospital (Sydney) ICU, John Flynn Private Hospital ICU, Maroondah Hospital ICU, The Townsville Hospital ICU, Ipswich Hospital ICU, Sir Charles Gairdner Hospital ICU, Manly Hospital, St Vincent's Private Hospital (Sydney) ICU, St Andrew's War Memorial Hospital ICU, Tamworth Base Hospital ICU, John Fawkner Hospital ICU, Westmead Private Hospital ICU, St Vincent's Hospital (Melbourne) ICU, Calvary Hospital (Canberra) ICU, Latrobe Regional Hospital ICU, Frankston Hospital ICU, The Memorial Hospital (Adelaide) ICU, St Andrew's Hospital Toowoomba ICU, Bundaberg Base Hospital ICU, Mount Hospital ICU, Royal Melbourne Hospital ICU, The Wesley Hospital ICU, Shoalhaven Hospital ICU, Calvary North Adelaide Hospital ICU, Grafton Base Hospital ICU, Melbourne Private Hospital ICU, St Vincent's Hospital (Toowoomba) ICU, Footscray Hospital ICU, Epworth Freemasons Hospital ICU, Mater Health Services North Queensland ICU, St John Of God Hospital (Geelong) ICU, Caboolture Hospital ICU, Dubbo Base Hospital ICU, Holy Spirit Northside Hospital ICU, Campbelltown Hospital ICU, Mildura Base Hospital ICU, Central Gippsland Health Service ICU, Goulburn Base Hospital ICU, Queen Elizabeth II Jubilee Hospital ICU, Manning Rural Referral Hospital ICU, Flinders Private Hospital ICU, Wimmera Health Care Group (Horsham) ICU, Norwest Private Hospital ICU, Hollywood Private Hospital ICU, Calvary John James Hospital ICU, Princess Alexandra Hospital ICU, Hervey Bay Hospital ICU, St

John Of God Health Care (Subiaco) ICU, Gosford Private Hospital ICU, Kareena Private Hospital ICU, Joondalup Health Campus ICU, Western District Health Service (Hamilton) ICU, Griffith Base Hospital ICU, Bunbury Regional Hospital ICU, South West Healthcare (Warrnambool) ICU, Sunshine Hospital ICU, St Andrew's Private Hospital (Ipswich) ICU, Pindara Private Hospital ICU, Mount Isa Hospital ICU, St Vincent's Private Hospital Fitzroy ICU, Noosa Hospital ICU, St John of God Hospital (Bendigo) ICU, Women & Childrens Hospital, Newcastle Private Hospital ICU, Buderim Private Hospital ICU, Epworth Eastern Private Hospital ICU, Robina Hospital ICU, Wyong Hospital ICU, Macquarie University Private Hospital ICU, Rockingham General Hospital ICU, Armadale Health Service ICU, Peninsula Private Hospital ICU, National Capital Private Hospital ICU, Sunshine Coast University Private Hospital ICU, Maitland Hospital HDU/CCU, Sydney Southwest Private Hospital ICU, Fiona Stanley Hospital ICU, Hurstville Private Hospital ICU, Wollongong Private Hospital ICU, Western Private (VIC), St John of God Midland Public & Private ICU, Gold Coast Private Hospital ICU, The Chris O'Brien Lifehouse ICU, Epworth Geelong ICU, Holmesglen Private Hospital, Sunshine Coast University Hospital ICU St John of God (Berwick) ICU, Angliss Hospital ICU, Werribee Mercy Hospital ICU, Maitland Private Hospital ICU.

## Author Contributions

**Conceptualization:** Kelly Thompson, Naomi Hammond, Lucy Modra, David Pilcher.

**Data curation:** David Pilcher.

**Formal analysis:** Michael Bailey, Gary Low, David Pilcher.

**Funding acquisition:** Kelly Thompson.

**Methodology:** Kelly Thompson, Michael Bailey, Lucy Modra.

**Project administration:** Kelly Thompson.

**Resources:** David Pilcher.

**Supervision:** David Pilcher.

**Writing – original draft:** Kelly Thompson, Naomi Hammond, Michael Bailey, Jai Darvall, Lucy Modra, David Pilcher.

**Writing – review & editing:** Kelly Thompson, Naomi Hammond, Michael Bailey, Jai Darvall, Gary Low, Steven McGloughlin, Lucy Modra, David Pilcher.

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
