## [Decision Letter · Decision Letter 0]

19 Dec 2022

PONE-D-22-30745Sex differences in survival after intensive care unit treatment for sepsis: a cohort studyPLOS ONE

Dear Dr. Thompson,

Thank you for submitting your manuscript to PLOS ONE. After careful consideration, we feel that it has merit but does not fully meet PLOS ONE’s publication criteria as it currently stands. Therefore, we invite you to submit a revised version of the manuscript that addresses the points raised during the review process. Please submit your revised manuscript by Feb 02 2023 11:59PM. If you will need more time than this to complete your revisions, please reply to this message or contact the journal office at plosone@plos.org. Please include the following items when submitting your revised manuscript:

We look forward to receiving your revised manuscript.

Kind regards,

Chiara Lazzeri

Academic Editor

PLOS ONE

Journal Requirements:

-https://doi.org/10.1016/j.jinf.2022.04.035

-10.1001/jamaneurol.2018.0123

In your revision ensure you cite all your sources (including your own works), and quote or rephrase any duplicated text outside the methods section. Further consideration is dependent on these concerns being addressed.

"KT received funding for the work from the National Health and Medical Research Council of Australia"

6. Please upload a new copy of Figure 2 and 3 as the detail is not clear. Please follow the link for more information:

https://blogs.plos.org/plos/2019/06/looking-good-tips-for-creating-your-plos-figures-graphics/

https://blogs.plos.org/plos/2019/06/looking-good-tips-for-creating-your-plos-figures-graphics/

Reviewers' comments:

Reviewer's Responses to Questions

**Comments to the Author**

1. Is the manuscript technically sound, and do the data support the conclusions?

Reviewer #1: Yes

Reviewer #2: Yes

2. Has the statistical analysis been performed appropriately and rigorously? 

Reviewer #1: Yes

Reviewer #2: Yes

3. Have the authors made all data underlying the findings in their manuscript fully available?

Reviewer #1: Yes

Reviewer #2: Yes

4. Is the manuscript presented in an intelligible fashion and written in standard English?

Reviewer #1: Yes

Reviewer #2: Yes

5. Review Comments to the Author

Reviewer #1: The aim of this study is to explore the impact of sex on survival for septic patients admitted in ICU The primary outcome is time to death for up to 3-years. The study included 50,956 patients from January 1, 2018 to December 31, 2020. The data were retrieved from ANZIC. Patient’s characteristics were different between men and women. Men were older (mean age 66·5 vs 63·6 years), more commonly received mechanical ventilation (27·4% vs 24·7%) and renal replacement therapy (8·2% vs 6·8%).

The FU was up to 3 years and Cox proportional hazard models were used to estimate the effect of sex on time to death. Since patients’ characteristics were different according to sex, analysis were adjusted for age, diagnosis, acute illness severity, body mass index, chronic comorbidities, frailty, source of admission to ICU and to hospital, treatment limitation at ICU admission, socio-economic status, region of Australia, type of hospital (regional/rural, metropolitan, tertiary or private) and year of admission to ICU.

Overall women had better survival for up to three-years after ICU admission. This result was confirmed by sensitivity analyses focusing on patients who survived the hospital admission. The effect of sex was more important among patients older than 75y than for patients younger than 54.

Comments:

This is a very nice study, addressing a question with some controversy. The large cohort of included patients validate the results although some other cofounding factors not considered in the adjustment might have contributed to the difference. It is also important to confirm that the better early prognosis was sustained over time up to 3 years after ICU admission.

The manuscript is well written, easy to read, tables and figures are clear.

Fig 3: what is the ref for CFS? OR is worse for CFS 5-6 than for CSF 7 an-8? Please comment

The inclusion period was partly during the COVID-19 surge period. Did you analyse specifically the impact of sex in this population? See for exemple (Sex-specific treatment characteristics and 30-day mortality outcomes of critically ill COVID-19 patients over 70 years of age-results from the prospective COVIP study; Canadian Journal of Anaesthesia. 2022-08-09)

Reviewer #2: This retrospective study of more than 50,000 sepsis ICU admissions in Australia showed that male sex was associated with an increased risk of ICU admission and that men had worse survival rates for up to 3-years, compared to women. As the authors themselves pointed out these findings are similar to those from a recent systematic review including 71,850 patients from 12 studies that reported a survival advantage for women at one year (Antequera A et al, 2021). The new information derived from this study, however, is that there is a survival advantage for women that persists for up to three years after sepsis.

The present study also indicates that in all three age groups (<55 years, 55 to 74 years, ≥ 75 years), women had better survival than men from sepsis. As the authors also pointed out, a 2021 retrospective review of 12,321 ICU records in Boston reported similar overall findings of higher mortality due to sepsis in men compared to women overall, but no difference in short and longer-term survival for men and women aged over 50 years (Lin S et al, 2021). No explanation is provided for the differences in the results published by Lin et al compared to this study.

The authors postulated that greater protection in women may be attributed to the immune enhancing effects of estrogen, and male disadvantage may be due to the immunosuppressive properties of male androgens, particularly testosterone. In this regard, testosterone is common intermediate between males and females and it gets converted enzymatically to 17β-estradiol by Aromatase or to 5α-dihydrotestosterone (DHT) by 5α-reductase. Thus, DHT rather than testosterone might be the active immunosuppressive steroid.

Additionally, estrogen decrease with aging in women and clearly the majority of the women in this study were post-menopausal with low estrogen levels. It is therefore possible that post-menopausal women in this study were on HRT and thus they had better survival than men. However, hormonal status nor information about HRT was not provided in this study.

In men, the levels of DHT also decrease with aging and thus aged men should do better than women – this, however, was not the case. So, the reason for the difference between men and women in survival remains unclear. Could it be due to a difference in the estrogen to DHT ratio? Unfortunately, this information was also not available in this study. Furthermore, appropriate explanation/justification for the difference in present from those of Lin S et al 2021 need to be provided.

6. PLOS authors have the option to publish the peer review history of their article (what does this mean?). If published, this will include your full peer review and any attached files.

Reviewer #1: No

Reviewer #2: No

---

## [Author Response · Author response to Decision Letter 0]

2 Feb 2023

24 January 2023

Editor-in-Chief

PLOS ONE

Dear Editor-in-Chief

RE: PONE-D-22-30745: Sex differences in survival after intensive care unit treatment for sepsis: a cohort study

Thank you PLOS ONE Editors and Reviewers for your thoughtful review of our manuscript. Please find below our response to reviewer and academic editor feedback which we believe has considerably improved the quality of our manuscript. 

Response: The files have been updated in accordance with PLOS ONE’s style requirements. 

-https://doi.org/10.1016/j.jinf.2022.04.035

-10.1001/jamaneurol.2018.0123

In your revision ensure you cite all your sources (including your own works), and quote or rephrase any duplicated text outside the methods section. Further consideration is dependent on these concerns being addressed.

Response: As the corresponding author of the first of these papers, I have compared the manuscripts in Microsoft word and cannot see any significant overlapping text which would raise plagiarism concerns. My colleague David Pilcher, the senior author on this paper has done the same for the JAMA Neurology paper and has also found minimal overlapping text outside of the methods section. Please note, the description of the ANZICS APD in the methodology section is similar in all papers that are written using this bi-national registry. It is written in this way to comply with ANZICS requirements and is specifically to ensure consistency of description throughout all publications. 

"KT received funding for the work from the National Health and Medical Research Council of Australia"

Response: Thank you for updating the online submission form on our behalf. The manuscript funding section at the end of the paper has been updated in tracked changes accordingly to confirm that: "The funders had no role in study design, data collection and analysis, decision to publish, or preparation of the manuscript." 

Response: Thank you for updating this on our behalf. The data availability statement should be as follows:

“Deidentified individual patient data are collected from each hospital ICU under Quality Assurance Legislation and standing agreements with each State and Territory health department of Australia and the Ministry of Health of New Zealand. Data are collected primarily for the purpose of benchmarking ICU activity and performance. Secondary access and use of the data is restricted other than under specific agreements with researchers. Applications for access to data can be made to anzics.core@anzics.com.au”

Response: Please see the above response including the updated data availability statement.

6. Please upload a new copy of Figure 2 and 3 as the detail is not clear. Please follow the link for more information:

Response: A new copy of Figure 2 has been uploaded. Please note, Figure 3 has now been separated into 2 figures, (Figs 3 and 4) to improve readability and clarity of text in the figure. 

Response: A new section with Supporting information file captions has been included at the end of the manuscript. Please confirm whether this is appropriate? After reading the supporting information instructions I was unsure as to whether the journal requires the figures and tables in the Supporting Information file to be uploaded to the manuscript. 

Response: No changes to the reference list were made. 

Comments to the Author

Reviewer #1: 

Comments:

Fig 3: what is the ref for CFS? OR is worse for CFS 5-6 than for CSF 7 an-8? Please comment

Response: The reference group for the Clinical Frailty Score is ‘not frail’(the top HR reported). The most frail group (CFS >6) had the highest HR in both women and men. The HR at the bottom is for those with an unknown frailty score. Please let us know if further clarification is required? 

The inclusion period was partly during the COVID-19 surge period. Did you analyse specifically the impact of sex in this population? See for exemple (Sex-specific treatment characteristics and 30-day mortality outcomes of critically ill COVID-19 patients over 70 years of age-results from the prospective COVIP study; Canadian Journal of Anaesthesia. 2022-08-09)

Response: Thank you for this comment. We included patients with COVID but did not examine COVID as a subgroup as this group represented a small component of the whole study population due to relatively few COVID admissions to ICU in Australia in 2020.

On re-examining the data, there were 990 probable COVID positive patients included (which includes some probable COVID positive patients from right at the beginning of the pandemic who didn’t get definitive tests on admission but were likely positive). Of these 337 were female (9.8% mortality) and 613 male (16.3% mortality). There were 456 confirmed COVID positive pneumonitis patients. Of these 163 were female (11.0% mortality) and 293 were male (15.7% mortality). We have now made a comment on this in the main manuscript and have reported additional information related to SARS-CoV-2 survival for females and males in the Supporting Information (Figs S3 and S4). 

Reviewer #2: 

The authors postulated that greater protection in women may be attributed to the immune enhancing effects of estrogen, and male disadvantage may be due to the immunosuppressive properties of male androgens, particularly testosterone. In this regard, testosterone is common intermediate between males and females and it gets converted enzymatically to 17β-estradiol by Aromatase or to 5α-dihydrotestosterone (DHT) by 5α-reductase. Thus, DHT rather than testosterone might be the active immunosuppressive steroid.

Additionally, estrogen decrease with aging in women and clearly the majority of the women in this study were post-menopausal with low estrogen levels. It is therefore possible that post-menopausal women in this study were on HRT and thus they had better survival than men. However, hormonal status nor information about HRT was not provided in this study. In men, the levels of DHT also decrease with aging and thus aged men should do better than women – this, however, was not the case. So, the reason for the difference between men and women in survival remains unclear. Could it be due to a difference in the estrogen to DHT ratio? Unfortunately, this information was also not available in this study. Furthermore, appropriate explanation/justification for the difference in present from those of Lin S et al 2021 need to be provided.

Response: As the reviewer points out, we did not have any data related to women’s post-menopausal status (beyond age) or their use of Hormone Replacement Therapy. We agree that it is possible that 5α-dihydrotestosterone (DHT) may have had a role in male immunosuppression and have added this in the discussion section. 

In regards to the different findings in our study compared to the study conducted by Lin et al in Boston, one possibility could be the difference in follow-up time. We have added this in tracked changes to our discussion section.

Thank you for your reconsideration of the manuscript for publication in PLOS ONE. 

Kind Regards

Dr Kelly Thompson

Conjoint Senior Lecturer, UNSW

Research Fellow, The George Institute for Global Health

---

## [Editor Report · Decision Letter 1]

5 Feb 2023

Sex differences in long-term survival after intensive care unit treatment for sepsis: a cohort study

PONE-D-22-30745R1

Dear Dr. Thompson,

We’re pleased to inform you that your manuscript has been judged scientifically suitable for publication and will be formally accepted for publication once it meets all outstanding technical requirements.

Kind regards,

Chiara Lazzeri

Academic Editor

PLOS ONE
---

## [Editor Report · Acceptance letter]

14 Feb 2023

PONE-D-22-30745R1 

Sex differences in long-term survival after intensive care unit treatment for sepsis: a cohort study 

Dear Dr. Thompson:

I'm pleased to inform you that your manuscript has been deemed suitable for publication in PLOS ONE. Congratulations! Your manuscript is now with our production department. 

Kind regards, 

on behalf of

Dr. Chiara Lazzeri 

Academic Editor

PLOS ONE